

# Trend analysis by a piecewise linear regression model
# applied to surface air temperatures in Southeastern Spain
# (1973-2014)
Pablo Campra[1], Maria Morales[2]
[1]Engineering School D2.36, University of Almeria, Almeria 04120, Spain
*Correspondence to:* Pablo Campra, pcampra@ual.es
[2]Mathematics Department. University of Almeria, Almeria 04120, Spain



**Abstract:** The magnitude of the trends of environmental and climatic changes is mostly derived from the slopes of the linear trends using ordinary least-square fitting. An alternative flexible fitting model, piecewise regression, has been applied here to surface air temperature records in southeastern Spain for the recent warming period (1973-2014) to gain accuracy in the description of the inner structure of change, dividing the time series into linear segments with different slopes. Breakpoint years, with confidence intervals (CIs), were estimated and separated periods of significant trend change were determined. First, simple linear trends for mean, maximum and minimum surface air temperatures and diurnal temperature range (DTR) from the four longest and most reliable historic records in SE Spain were estimated. All series in the region showed intense linear warming signs during the period 1973-2014. However, updated warming trends were lower than those previously cited for the region and Spain from the 1970s onwards. Piecewise regression model allowed us to detect breakpoints in the series, and the absence of significant trends in the most recent period of the segmented fits for two stations. In general, piecewise regression model showed better fit than simple linear regression model, and thus, showed a better description of temperature variability.

**Keywords:** temperature trends; piecewise regression; segmented regression; Southeastern Spain; regional climate change; warming slow-down;



## 1. Introduction

Trend analysis is useful for a better understanding of climate change and variability. The estimation of simple linear trends is the most straightforward assessment of the long-term behavior of a time series in climate change studies. However, due to changes in the trends, real-world time series do not generally fit in a straight line. Thus, simple linear regression may be misleading, as it does not describe the inner structure of change in the series, ignoring the existence of significant changes in the slope of the linear fit, called breakpoints. Such simple linear trends may be illogical and physically meaningless for climatic data analyses, where the linearly fitted trend makes little sense, for the underlying mechanisms of global climate change are likely to be nonlinear and nonstationary, so other methods of time series analysis might be advisable (Seidel and Lanzante, 2004; Wu et al, 2007; Sefidmadgi et al, 2014). In particular, linear trend does not adequately describe low-frequency behavior of temperature time series (Karl et al., 2000). Piecewise regression model fits a nonlinear function with a non constant rate of change, and has been applied to analyze time series of different climatic variables to detect breakpoints in linear trends. Karl *et al*, (2000), identified the timing of change points in global temperature time series by minimizing the residual sum of squares of all possible combinations of four line segments representing time intervals of 15 years or more. Tome and Miranda (2004) adapted that fitting method to develope an algorithm for fitting a continuous regression model with several break points to data and then it was applied local changes in temperature, precipitations and the NAO index in Portugal. Liu et al. (2010) used the same method to find partial trends of wind variability in the mesosphere and the lower thermosphere over a local observatory at Collm, Germany.

Piecewise regression is a method of regression analysis where the response variable is split in two or more intervals, and a line segment is fitted to each interval, with the constraint that the regression function will be continuous. Each line is connected at an unknown value called breakpoint. Piecewise regression is suitable for situations where the response variable shows abrupt changes within a few values of the explanatory variable, (Toms and Lesperance, 2003). This flexible regression method is scarcely used in the analysis of long term trends of climatic variables, though in many cases it offers a better fit to the records, and shows better compliment with the assumptions of regression analysis.

Here we have applied to regional temperature records from SE Spain an alternative fitting approach for long term climatic series, piecewise or segmented regression. SE Spain includes some of the most semi-arid areas in the Northern Mediterranean, with increased vulnerability to projected shifts in global circulation and pressure patterns, so it is of great interest the characterization and update of long-term trends in this area. Climate change projections for the Mediterranean region show that high-temperature conditions are generally expected to increase in the future (Giorgi and Lionello, 2008; Jacobeit et al., 2014). Over the Iberian Peninsula, significant high rates of surface air temperature warming have been recorded by the Spanish Meteorological Agency (AEMET) from the early 1970s onward, in accordance with the last period of accelerated global warming (Hartmann *et al*., 2013). For instance, Brunet et al.



(2007) estimated recent average rates of warming for Spain during 1973-2005 of 0.48 ℃ decade[-1] for
mean temperatures ($T_{mean}$), with 0.51 and 0.47 ℃ decade[-1] for maximum ($T_{max}$), minimum ($T_{min}$)
temperatures respectively. Del Rio *et al*. (2012) reported rates of around 0.3 °C decade[-1] in over 90% of
Spanish weather stations for the period 1961-2006. However, most studies just have a limited spatial or
temporal coverage, and a scarce number of studies have estimated recent changes in long-term mean
annual temperature trends in Spain from the early 1970s into the 2010s (El Kenawy et al., 2012; Turco et
al., 2014; Gonzalez-Hidalgo et al., 2015). In particular, assessments of observed changes in SE Spain are
scarce. Fernández-Montes and Rodrigo (2015) have recently reported temperature trends in SE Spain for
the period 1970-2007, estimating increases in $T_{min}$ and $T_{max}$ in most of the region of 1.2 and 0.5 ℃
decade[-1] respectively. However, no update of SE Spain temperature trends including recent 2010s years
has been reported to date.
Long-term global warming has been unequivocal from mid 1970s to 2013, with an average global trend
of 0.2 ℃ decade[-1] (Hansen *et al*., 2010; Lawrimore *et al*., 2011; Jones et al., 2012; Rohde *et al*., 2013;
Turco et al., 2015). However, recent warming trends throughout the planet have either decreased or lost
statistical significance in many regional series in the planet from around 1997 (Easterling and Wehner,
2009; Kaufmann et al., 2011), though temperatures still remain well above the long-term average. Recent
observations and global averages show a significant decrease in the warming trend from 0.12 °C decade[-1]
in the period 1951-2012 to 0.05 °C decade[-1] in 1998-2012 (Hartmann et al., 2013). Nevertheless, such
slowing for a decade or so has been seen in past observations and has been properly simulated in climate
models (Met Office, 2013). Surface air temperatures are characterized by wide spatial and temporal
variability (Lovejoy, 2014; Steinmann *et al*., 2015, Turco *et al*., 2015), and there is a need to determine
local and sub-regional trends and variability to gain knowledge of global climate change patterns.
A simple eye inspection of the main temperature records in SE Spain suggest a possible breakpoint year
at every series from the early 1990s (Fig. 1), showing decreased rates of warming since then. In order to
detect these breakpoints we have used a piecewise regression approach, estimating the continuous set of
straight lines that better fits every time series of annual $T_{mean}$, $T_{max}$, $T_{min}$, and diurnal temperature range
(DTR), deriving confidence intervals (CIs) for the breakpoint estimates. The main goals of this work
were: first use a simple linear regression model to update the long-term warming trends of air surface
temperatures at AEMET first order stations in SE Spain from the year 1973 to 2014; and second, to test
the goodness of fit and prediction skills of a piecewise regression model compared to simple linear
regression applied to the same period. Our main interest in this work is to discuss if there is a recent
significant breakpoint in long-term warming trends in SE Spain, and whether it is statistically consistent
with the long term warming, likely to happen superimposed on the longer-term warming trend, or whether
the last records might represent a key emerging change-point in long term trends at the area.

**2. Data temperature series**



We have selected four meteorological stations with the longest, continuous and most reliable records in
SE Spain (Table 1): Almeria (AL), Granada (GR), Malaga (MA), and Murcia-San Javier (MU). These
stations are well-spaced across SE Spain (Fig. S1), with a minimum linear distance of 90 Km (between
GR and MA), and maximum of 340 km between MA and MU, below the threshold of 400 km that has
been suggested as optimal for building a representative meteorological network in Spain (Peña-Angulo et
al., 2014). AL, MA and MU are representative of coastal Mediterranean climate, while GR shows a
"continentalized" inland Mediterranean climate, with higher extremes of warm and cold days in summer
and winter, respectively. Their records cover at least from 1973 to 2014, including the recent period of
accelerated global warming from the 1970s that we intended to analyze in our area of study. These
stations are located within international airports, and belong to the first order (synoptic) network of the
Spanish official meteorological agency (AEMET). This selection was based on potential data quality
from highly monitored sites and good-quality records controlled by the *Servicio de Desarrollos*
*Climatológicos* (SDC, Climatological Branch) of AEMET. Raw data of daily $T_{max}$ and $T_{min}$ have
undergone quality control checks to avoid syntax, internal consistency and temporal coherence errors, and
controls of extreme thresholds and spatial coherency. Additionally, these records have been extensively
analyzed for artificial in-homogeneities in previous studies (Brunet et al., 2008; Staudt et al., 2007).
**3. Regression methods**
**3.1. Simple linear regression**
As a first conventional approach to detect temperature change, simple linear regression was applied to the
series. Trends and their 95% CIs were estimated by least squares linear regression.  Linear trends were
estimated in every series from the slopes of the fit using values of annual averages of $T_{max}$, $T_{min}$, $T_{mean}$ and
DTR, calculated from monthly means provided by AEMET.

**3.2. Piecewise regression**

However, when the residuals of simple linear fits for each $T_{mean}$ series were tested for the assumptions of
normality, independence, homoscedasticity and linearity, it turned out that: a) homoscedasticity was is not
met by GR station; b) the linear assumption was not verified by AL and MU residuals; c) the
independence assumption was rejected by Ljun-Box in AL. The violation of the homogeneous variance
assumption could result in unreliable estimates of the standard errors that might turn out in mistaken
conclusions over the slope. In these cases, heteroscedasticity-consistent (MacKinnon and White, 1985),
and autocorrelation-consistent estimators have been used (Newey and West, 1987). To solve these
problems, here we have tested and alternative regression model, piecewise regression. We have used a
segmented model between the mean response E(Y) and the explanatory variable Z, modeled by adding in
the linear predictor the terms. Eq. (1):





1                       $\beta_1 z_i + \beta_2 (z_i - \psi)_+$         (1)

where $\beta_1$ is the slope of the left line segmented, $\beta_2$ is the 'difference-in-slopes', $\psi$ is the breakpoint, and
$(z_i - \psi)_+ = (z_i - \psi) \times I(z_i > \psi)$ being I(A)=1 if A is true. In order to estimate the break-points
location, we use the approach suggested by Muggeo (2003) and at the R package 'segmented' (Muggeo,
2008). Karl *et al*. (2000) used this approach to obtain a better fit of global temperatures than simple linear
regression. Tomé and Miranda (2004) further developed an algorithm to identify best location for
breakpoints in climatic series. Here we have applied a piecewise regression model to those series where
there is enough statistical evidence to support the existence of breakpoints. Smoothed scatter plots were
used to provide the starting values for breakpoints in order to improve the convergence of the algorithm,
and we have checked the existence of a significant breakpoint by testing over the difference in slope
(Muggeo, 2003). We have employed the R package to estimate the parameters of the piecewise
regression in a deterministic way, and to fit linear segments to the data. The analysis of the residuals from
piecewise regression fits of our data showed that the assumptions of normality (Shapiro-Wilks and
Anderson-Darling tests), independence (Ljung-Box test) and homocedasticity (Breusch-Pagan test) were
met by all the series, with the exception of GR. In this case, the robust variance estimator proposed by
MacKinnon and White (1985) was used. In order to test for a significant slope, we have applied a Wald's
test. As general criteria, we have not estimated the slope of segments when they represented time
intervals of less than 5 years. As stated by Tome and Miranda (2005), ", if the first or the last breakpoint
happen near the minimum allowed position the result should be looked upon with some suspicion". These
graphs are not shown in the figures at the results section. hh
**3.3. Evaluation of regression models**
We have compared both fitting and predicting performance of piecewise and simple linear models applied
to the whole length of data available for every series of $T_{mean}$, $T_{max}$ , $T_{min}$ and DTR (Table 2). In order to
compare the goodness-of-fit of the models, we calculated $R^2$ for each fit, and the residual standard error
(RSE) in order to avoid the artificial skill of $R^2$. Also, we carried out a cross-validation analysis (Hastie et
al., 2009) to compare the goodness-of-forecasting skills among models. Thus, the data were divided into 5
roughly equal-sized parts, and for the $i_{th}$ part, i=1,…,5, the model was fitted using the other 4 parts of the
data. The prediction error of the fitted model was estimated when predicting the $i_{th}$ part of the data.
Finally, the mean square error ($MSE_{CV}$) of the 5 evaluation parts was calculated as forecasting skill
indicator. As we can observe in Table 2, piecewise regression showed a superior behavior both fitting and
forecasting compared to simple linear regression.

**4. Results and discussion**



A casual eye inspection of SE Spain temperature series for 1973-2014 suggests two periods of different
change rates (Fig. 1): a first period of intense warming, from 1973 until the early 1990s; and a second
period of lower or not significant rates of temperature increase from 1989 to 2014. The sharp drop in
1991, associated to Mt. Pinatubo eruption, was transient and widespread throughout the planet. As a first
approach, simple linear regression slopes for 1973-2014 showed average warming trends of $T_{mean}$ in the
range of 0.25-0.46 ºC decade$^{-1}$ (Table 3), below average rates cited in previous studies for this area and
for Spain (Brunet *et al*., 2007; Fernandez-Montes and Rodrigo, 2015). However, these studies don't cover
the most recent records of the 2010s, and they are difficult to compare with our results because of the
different areas, data sets and periods used (Brunet et al., 2007; Del Rio *et al*., 2012; Gonzalez-Hidalgo et
al., 2015). In all stations except GR, mean annual DTR decreased at significant rates, between -0.25 and-
0.36 ºC decade$^{-1}$, due to higher increases of $T_{min}$ (0.43/0.58 ºC decade$^{-1}$) than $T_{max}$. This trend in DTR is in
agreement with the expected "fingerprint" for global greenhouse warming (Hartmann *et al*, 2013), but is
opposite to the positive trend in annual DTR reported for Spain for the last decades (Galan *et al*., 2001;
Brunet *et al*., 2007; Del Rio *et al*., 2012). However, DTR reductions have been cited in Southern parts of
Spain (Horcas *et al*., 2001; Staudt *et al*., 2005; Gonzalez-Hidalgo *et al*., 2015) and have been related to
$T_{min}$ increases driven by land-use changes. The physical interpretation of DTR trends, and the
contribution of local forcings due to land-use changes still remain a key uncertainty, and confidence in
this indicator has been defined as *medium* in reported decreases in observed DTR (Hartmann et al.*,* 2013).
Spatial variability of trends in $T_{min}$ and $T_{max}$ was high, as described by with Peña-Angulo *et al*. (2014),
who detected in SE coastal Mediterranean areas the highest variability for mainland Spain, and again
suggested an association with the dramatic land-use changes by urbanization and agriculture at this coast
in the last decades. However, the two more distant stations, MA and MU, two coastal stations that lie 340
km away, showed comparable patterns of change for all variables, different to trends in GR and AL, with
higher rates of increase in all variables, and $T_{min}$ rates of increase of 0.5 ºC decade$^{-1}$. In GR no significant
change in long-term DTR was detected, with only slight differences in rates of warming for $T_{max}$ and $T_{min}$,
in agreement with described patterns from 1970s for Spanish southern plateau (Galan *et al*., 2001), with a
continentalized Mediterranean climate that is more similar to GR climate than to coastal stations. AL
showed the lowest rates of warming in the area, probably related to the dramatic land-cover change
towards greenhouse farming developed from the 1980s, resulting in the widest concentration of high
albedo greenhouses in the planet (Fig. S2). The alteration of the local energy balance by widespread
reflectivity increase has probably offset in this area the global forcing exerted by the increase of
greenhouse gases (Campra *et al*., 2008; Campra and Millstein, 2013). DTR decrease due to higher rates of
warming of $T_{min}$ than $T_{max}$ has been also associated to urbanization and land cover changes (Mohan and
Kandya, 2015). MA is the only station in this study that might have been affected by the rapid growth of
urban areas (Malaga city), and the recent growth of low albedo pavements around it by recent
development of taxiways and runways from 2005 (Fig. S3) (*Fomento* Ministry of Spain, 2007). This
recent land cover change around the station might have enhanced warming trends in this record,
increasing heat retention properties of the surface (McKitrick and Michaels, 2007). In GR and MU, little
increase in urbanization has been produced around the stations, located in international airports far from
the growing cities, so trends in temperatures and DTR cannot be clearly associated here to local land
cover changes. Besides these local factors, the influence of the Mediterranean is a key forcing of SE
Spain temperature records. Observations of Mediterranean surface temperature indicate strong warming
from 1973. The average rate of warming at Western Mediterranean was 0.22 °C decade$^{-1}$ over 1973-2008
(Skirlis $et$ $al$., 2012). The Alboran Sea surrounding SE Spain showed an averaged warming trend of 0.31
C decade$^{-1}$ on 1982-2012 (Shaltout and Omstedt, 2014).
Alternatively, piecewise regression model from the full historic records available at each station offered a
better fit to the data in all temperature series (Fig. 2), showing that this model represents the observations
more accurately than the simple linear regression model generally used to describe climatic trends. As a
general criteria, in figures 2, 3 and 4 we have not represented piecewise regressions where last segment
are less than 5 years long. Uncertainty analysis of piecewise regression fits for every series and variable
are given in Tables S1-S4. This flexible regression model allowed for detection of several trend change
points, located in different years for every series and for every variable (Table 4). Recent $T_{mean}$ trends
were not significant from these breakpoints in AL and GR. In MU, $T_{mean}$ increase was less intense since
the 1982 breakpoint (Table 5). On the contrary, a breakpoint in 2013 was found in $T_{mean}$ in MA driven by
the historic $T_{mean}$ record in 2014 (Table 4).
In order to identify the components of change of $T_{mean}$, we have carried out additional piecewise
regression fits to $T_{max}$ and $T_{min}$ (Table 6). Breakpoints in both $T_{max}$ and $T_{min}$ were detected for all series
except MA (Fig. 3). DTR showed decreasing trends in the last segments of AL and MU (Fig. 4), but
recent breakpoints towards increasing DTR have appeared in MA (2012) and GR (2011), though only
significant at 10% (figures not shown) (Table 6). However, we found no signs in our data of the pattern of
change in described in global DTR, characterized by a decline and subsequent increase from a breakpoint
in mid 1980s, associated to global dimming and subsequent brightening (Wild $et$ $al$., 2007). (Table 6). In
AL, the 1989 breakpoint in $T_{mean}$ was driven by a significant fall of $T_{max}$ from 1987, further maintained by
the later stabilization of $T_{min}$ (with no significant trend from 1998). As a consequence of these trends, AL
station showed the highest recent rates of DTR reduction in the area (-0.61 °C decade$^{-1}$ since 1982),
probably due to local land cover changes stated before. In GR, the 1997 breakpoint in $T_{mean}$ was driven by
stabilization of $T_{min}$, with no significant trend from 1997. In MU a breakpoint was located around the
early 1980s in both $T_{max}$ and $T_{min}$, coincident with $T_{mean}$ breakpoint. No significant breakpoints in $T_{max}$ and
$T_{min}$ were detected in MA, showing a continuous warming trend during the period of study, along with the
highest simple linear rates in the region (+0.46 °C decade$^{-1}$). MA is the only series that shows no clear
signs of warming slow-down in $T_{min}$ and $T_{max}$ in recent years. Furthermore, a 2013 breakpoint was
detected in $T_{mean}$, due to 2014 breaking record, increasing long-term rates of warming. A similar 2013
breakpoint was shown by $T_{max}$ in GR. Our regression analyses does not allow us to suggest that these
recent breakpoints are signs of a future increase in warming trends in these stations, and this might lead to
failed conclusions when piecewise segments length is not long enough (Karl $et$ $al$, 2000).



## 5. Conclusions

Simple linear trends of temperatures at SE Spain for the period 1973-2014 reported here are consistent with the magnitude of warming described previously for the region. However, our results show that the generalized use of simple linear regression fits for the estimation of long term trends might not be sufficiently accurate to describe the structure of change of temperatures in long-term series, while flexible models such as piecewise regression provide better fits, and allows the detection of key breakpoints in the trends. Besides, the method is also much simpler and easier to interpret than most non-linear function fitting. By flexible regression, we have detected a recent slow-down in long-term warming trends in three of the four main records in SE Spain, with no significant trends in $T_{mean}$ in the most recent segments of two of them (AL and GR). No matter the uncertainty in the exact location of breakpoints reported here, given by 95% CIs (Table 4), we have shown statistical evidence of the presence of such reduction in warming rates in the area, consistent with previous reports of recent decreasing warming trends in global series. However, and given the limited length of the observations, we cannot suggest by our analysis that the absence of statistically significant warming in the most recent segments of piecewise regression model are necessarily inconsistent with historic variability of the series studied. Furthermore, recent breakpoints observed in MA and GR, might be signing increasing warming trends in the future, enhanced by 2014 global historic record (Jones, 2015), year-to-date global temperatures (up to June 2015) (NOAA, 2015a). Spatial differences in long-term trends and breakpoints location might be due to undetermined local dynamics in climate and/or forcings that still remain to be determined in every station, or just simply due to natural variability. To obtain statistical evidence about possible causes of the slow-down in warming rates in the area, future investigations should search for co-breaks and overlapping CIs in the break dates detected at the historic series of key local forcings (Estrada *et al*, 2013), helping to explain the observed non-linear trends, and the location of the trend changing points in the temperature series described here. It must also be taken into account that the method of piecewise regression is strongly dependent on the first guesses and imposed parameters such as minimum time distance between breakpoints, and minimum trend change between breakpoints (Tome and Miranda, 2005). Complementary to this search of statistical relationships, recent simulation studies with optimized treatment of global climate forcings have shown better insight to possible causes of a possible slow-down in global warming (Smith *et al*., 2014).

*Author contribution.* P. Campra designed the scope of the research, performed the simple linear regressions and wrote the manuscript, and M. Morales carried out all deep statistical analyses, residuals tests and developed the piecewise regression model in R.

*Acknowledgements:* We thank the Ministry of Economy and Competitiveness (MINECO) of the Spanish Government, and FEDER Funds from European Union for the Grant CGL2013-46873-R that has supported this work. The authors would like to acknowledge AEMET (Spanish Meteorological Agency) for providing data used in this work.





**Supporting Information**
Map of location of stations in SE Spain, picture of greenhouses land cover around AL station, picture of
MA airport new taxiways. Tables of uncertainty estimates of piecewise regression of mean, maximum,
minimum and DTR temperature series. This material is available free of charge via the Internet at …




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

| | ALMERIA (AL) | GRANADA (GR) | MALAGA (MA) | MURCIA (MU) |
|---|---|---|---|---|
| Location | 36° 50' 47" N | 37° 11' 23" N | 36° 39' 58" N | 37° 47' 20" N |
| | 2° 21' 25" W | 3° 47' 22" W | 4° 28' 56" W | 0° 48' 12" W |
| Altitude (msl.) | 21 | 567 | 5 | 4 |
| Length* | 1972-2015 | 1973-2015 | 1950-2015 | 1946-2015 |



1    Table 2. Goodness-of-fit test ($R^2$/RSE), and predictive value ($MSE_{CV}$) of the two regression models for $T_{mean}$,

2    $T_{max}$, $T_{min}$ y DTR. $R^2$ = coefficient of determination; RSE= residual standard error; $MSE_{CV}$ = mean square

3    error for cross-validation.

| | | PIECEWISE | | | SIMPLE LINEAR | | |
|---|---|---|---|---|---|---|---|
| | | $R^2$ | RSE | $MSE_{CV}$ | $R^2$ | RSE | $MSE_{CV}$ |
| $T_{mean}$ | AL | 0.60 | 0.32 | 0.14 | 0.40 | 0.39 | 0.14 |
| | GR | 0.42 | 0.53 | 0.29 | 0.36 | 0.55 | 0.32 |
| | MA | 0.76 | 0.33 | 0.12 | 0.74 | 0.34 | 0.12 |
| | MU | 0.67 | 0.38 | 0.15 | 0.59 | 0.42 | 0.19 |
| $T_{max}$ | AL | 0.52 | 0.37 | 0.17 | 0.03 | 0.52 | 0.27 |
| | GR | 0.39 | 0.64 | 0.32 | 0.28 | 0.68 | 0.33 |
| | MU | 0.57 | 0.31 | 0.10 | 0.49 | 0.33 | 0.12 |
| $T_{min}$ | AL | 0.70 | 0.36 | 0.17 | 0.67 | 0.38 | 0.27 |
| | GR | 0.40 | 0.58 | 0.38 | 0.29 | 0.62 | 0.45 |
| | MA | 0.79 | 0.38 | 0.30 | 0.79 | 0.37 | 0.26 |
| | MU | 0.66 | 0.53 | 0.28 | 0.58 | 0.58 | 0.38 |
| DTR | AL | 0.73 | 0.35 | 0.16 | 0.46 | 0.49 | 0.24 |
| | GR | 0.21 | 0.62 | 0.38 | 0 | 0.69 | 0.47 |
| | MA | 0.53 | 0.31 | 0.11 | 0.49 | 0.32 | 0.11 |
| | MU | 0.46 | 0.42 | 0.18 | 0.41 | 0.43 | 0.20 |



**Table 3. Change in annual mean (T$_{mean}$), maximum (T$_{max}$), minimum temperatures (T$_{min}$), and diurnal**
**temperature range (DTR), estimated as the slope of a simple linear regression fit (in ºC decade$^{-1}$), and**
**associated 95% confidence intervals (CI) for the recent warming period 1973-2014, at SE Spain first order**
**AEMET stations. Not significant values in italics (p>0.05).**

| Station | Tª series | Decadal coefficient (ºC) | 95% CI |
|---------|-----------|--------------------------|--------|
| AL | **T$_{mean}$** | **0.25** | **(0.15/0.36)** |
| | T$_{max}$ | *0.07* | *(-0.07/0.2)* |
| | T$_{min}$ | 0.43 | (0.34/0.53) |
| | DTR | -0.36 | (-0.49/-0.24) |
| GR$^{*}$ | **T$_{mean}$** | **0.33** | **(0.19/0.48)** |
| | T$_{max}$ | 0.34 | (0.17/0.52) |
| | T$_{min}$ | 0.32 | (0.16/0.48) |
| | DTR | *-0.02* | *(-0.16/0.20)* |
| MA | **T$_{mean}$** | **0.40** | **(0.30/0.51)** |
| | T$_{max}$ | 0.26 | (0.18/0.35) |
| | T$_{min}$ | 0.55 | (0.40/0.70) |
| | DTR | -0.29 | (-0.4/-0.18) |
| MU | **T$_{mean}$** | **0.46** | **(0.37/0.55)** |
| | T$_{max}$ | 0.34 | (0.24/0.43) |
| | T$_{min}$ | 0.58 | (0.48/0.68) |
| | DTR | -0.25 | (-0.34/-0.17) |





1    **Table 4. Breakpoint estimates and 95% CIs for annual $T_{mean}$, $T_{max}$, $T_{min}$ and DTR series in SE Spain. Values in**

2    *italics* **are not significant at 5% level. $^{+}$ Significant at 10%**

| Station | $T_{mean}$ | $T_{max}$ | $T_{min}$ | DTR |
|---------|------------|-----------|-----------|-----|
| AL | $1989 \pm 5.3$ | $1987 \pm 3.4$ | $1998 \pm 11.4$ | $1982 \pm 2.6$ |
| GR | $1997 \pm 11.8$ | $2013 \pm 1.0$ | $1997 \pm 9.2$ | $2011 \pm 2.3$ |
| MU | $1982 \pm 5.7$ | $1983 \pm 7.2$ | $1981 \pm 5.0$ | $1981 \pm 4.1^{+}$ |
| MA | $2013 \pm 1.2^{+}$ | - | *$1977 \pm 10.7$* | $2012 \pm 3.2^{+}$ |



**Table 5. Trends in ºC decade$^{-1}$ of annual mean temperature change ($T_{mean}$) from simple linear regression fit**
**(SL) for each station (in bold), compared to the two successive sub-periods defined by piecewise (PW)**
**regression. Trend values in *italics* are not significant at 5% level. *Breakpoint was detected in 2013 and no**
**segmented regression periods are reported.**

| Station | Regression model | Period | Estimate | 95% CI |
|---|---|---|---|---|
| AL | **SL** | **1973-2014** | **0.25** | **(0.15/0.36)** |
|  | PW$_1$ | 1973-1989 | 0.76 | (0.51/1.0) |
|  | PW$_2$ | 1989-2014 | *-0.07* | *(-0.15/0.14)* |
|  |  |  |  |  |
| GR | **SL** | **1973-2014** | **0.33** | **(0.19/0.48)** |
|  | PW$_1$ | 1973-1997 | 0.55 | (0.34/0.76) |
|  | PW$_2$ | 1997-2014 | *-0.03* | *(-0.57/0.51)* |
|  |  |  |  |  |
| MU | **SL** | **1973-2014** | **0.40** | **(0.30/0.51)** |
|  | PW$_1$ | 1973-1982 | 1.28 | (0.69/1.87) |
|  | PW$_2$ | 1982-2014 | 0.27 | (0.14/0.4) |
|  |  |  |  |  |
| MA[*] | **SL** | **1973-2014** | **0.46** | **(0.37/0.55)** |



1    Table 6. Trends in ºC per decade of annual maximum ($T_{max}$) and minimum ($T_{min}$) temperature anomalies and

2    diurnal temperature range (DTR) for the periods defined by piecewise ($PW_{1-3}$) regression model. Trend values

3    in *italics* are not significant at 5% level. [+] Significant at 10%

| Station | | $T_{max}$ | $T_{min}$ | DTR |
|---------|---|-----------|-----------|-----|
| AL | **Breakpoint** | **1987** | **1998** | **1982** |
|    | $PW_1$ | 1.08 | 0.58 | 1.31 |
|    | $PW_2$ | -0.32 | *0.16* | -0.61 |
| GR | **Breakpoint** | **2013** | **1997** | **2011** |
|    | $PW_1$ | 0.29 | 0.63 | *-0.09* |
|    | $PW_2$ | - | *-0.18* | - |
| MU | **Breakpoints** | **1983** | **1981** | **1981**[+] |
|    | $PW_1$ | 0.82 | 1.92 | -1.01 |
|    | $PW_2$ | 0.16 | 0.38 | *-0.20* |
| MA | **Breakpoint** | **-** | **-** | **2012**[+] |
|    | $PW_1$ | - | - | -0.28 |
|    | $PW_2$ | - | - | - |





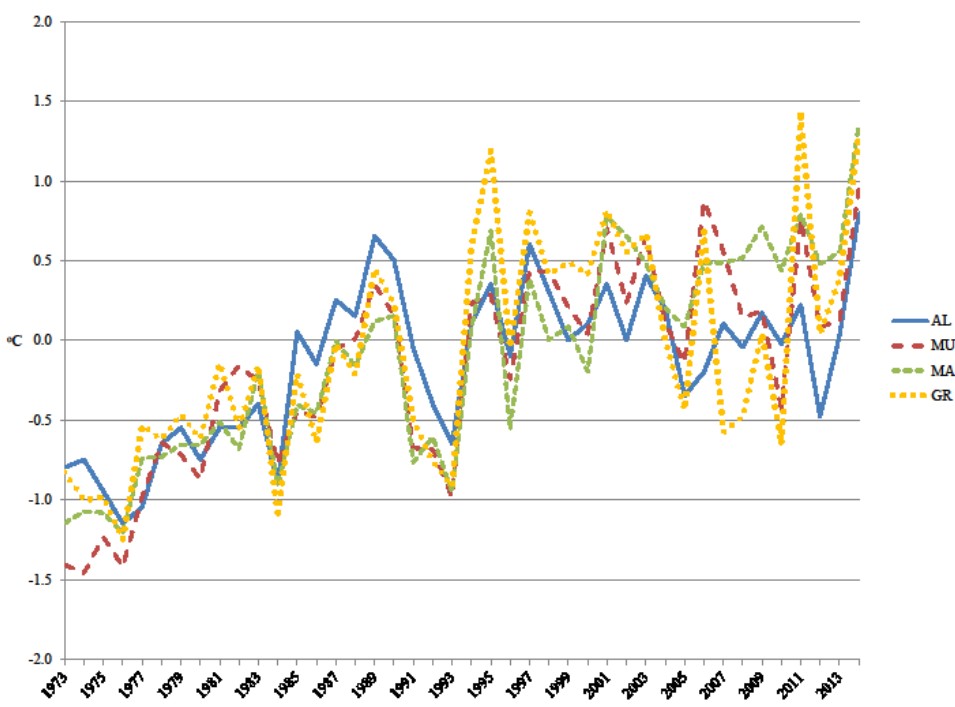

3    **Figure 1. Mean annual temperature anomaly series (ºC) from 1973-2014 at SE Spain First order stations of the**

4    **Spanish official meteorological network (AEMET). Reference period: 1981-2010. AL=Almeria; GR=Granada;**

5    **MA=Malaga; MU=Murcia**



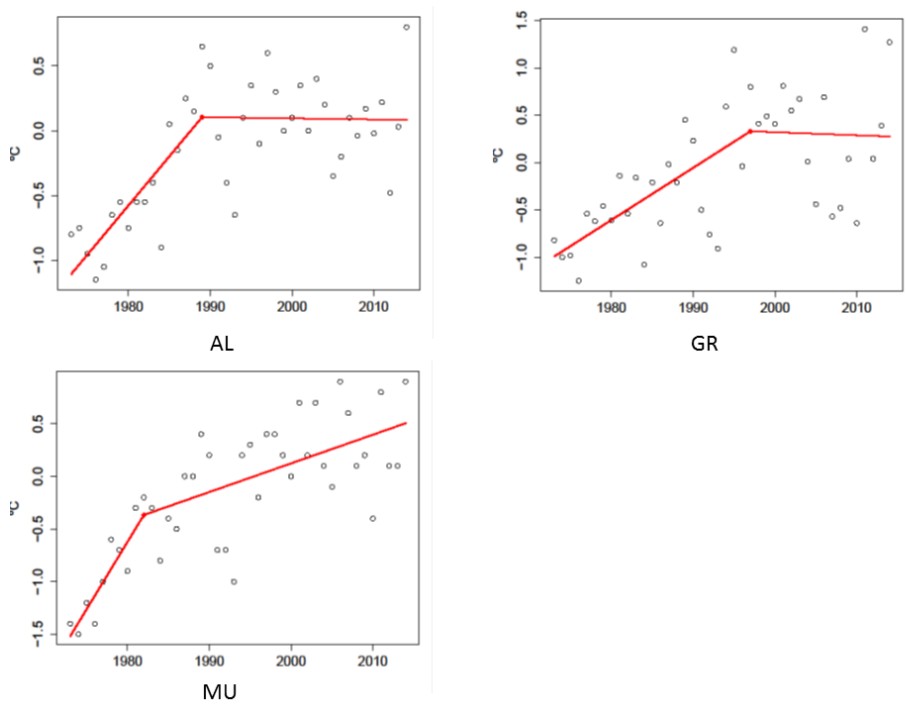

2      **Figure 2. Piecewise regression fitting of historic records of mean annual temperature anomalies (ºC) in SE**

3      **Spain. Only those series with last segment > 5 years are shown. AL=Almeria; GR=Granada; MU=Murcia.**





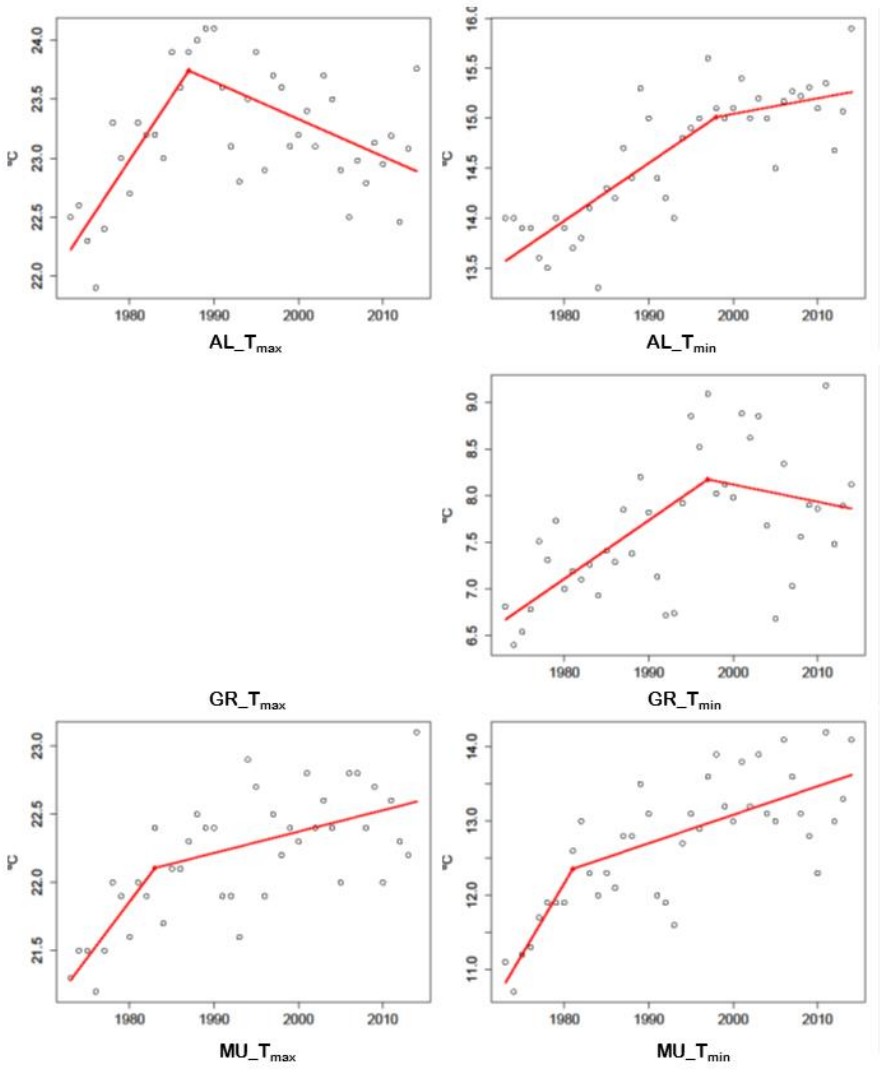

**Figure 3. Piecewise regression fitting of historic records of annual average maximum (Tmax) and minimum**

**(Tmin) temperature anomalies (ºC) in SE Spain stations. Only those series with last segment > 5 years are**

**shown. AL=Almeria; GR=Granada; MU=Murcia.**

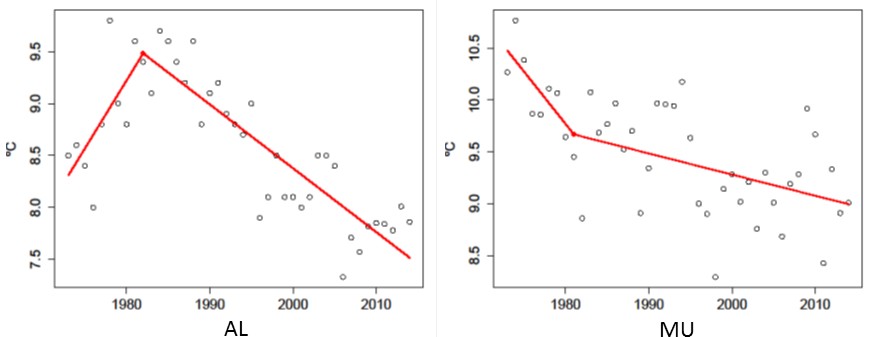

3  **Figure 4. Piecewise regression fitting of historic records of diurnal temperature range (DTR) in ºC in SE Spain**

4  **stations. Only those series with last segment > 5 years are shown.  AL=Almeria; MU=Murcia. (MA and GR**

5  **series not shown)**

