# Peer review of "Nonlin. Processes Geophys. Discuss., doi:10.5194/npg-2016-29, 2016 Manuscript under review for journal Nonlin. Processes Geophys. Published: 30 May 2016 © Author(s) 2016. CC-BY 3.0 License."

_Nonlinear Processes in Geophysics, 2016_

## Referee Comment (RC1) · Anonymous Referee #1 · 20 Jun 2016

The paper applies a published algorithm (Muggeo, 2003) to datasets of observations in SE Spain. There seems to be no innovation in the methods. In what concerns the analysis of results, I generally found it unconvincing, for reasons explained below.

The recent deceleration of global warming is a global process, not specifically linked to anything peculiar of SE Spain. That fact makes the discussion of the location of the breakpointinthe4chosenstations, and the slight observed differences between them, rather irrelevant. Attributing the global deceleration to specific land use changes in a very localized region is certainly wrong, and it seems to implied by the text (although not directly).

So, while I believe that a global analysis of mean or of spatially distributed surface

temperature would probably find a breakpoint in the beginning of 2000s, a fact that comes from a visual inspection of mean temperature, I don't see the point of doing the same analysis on such a small set of data, unless you have something new in the methodology or in the attribution analysis. That does not seem to be the case.

Furthermore the so called "hiatus" seems to be over, after the record breaking temperatures in 2015, making the results a bit outdated.

Finally I found the discussion supported by Table 2-4 rather weak. The piecewise approach will always produce a better fit, the question being the statistical support for the extra degrees of freedom. The large differences found in the location of the breakpoints in nearby stations mostly makes me feel uncomfortable about the robustness of the individual results and not excited to look for the physical explanation of those differences.

---

## Author Comment (AC1) · 15 Jul 2016

Reviewer 1 (Rew1 now on) seems to have misunderstood the scope of this research, and some statements in the manuscript. We answer one by one to these questions:

-"There seems to be no innovation in the methods" As the title states, this is a statistical analysis of data to estimate trends of change, using both conventional linear regression and a non-linear regression. The innovation here relies in the use of a piecewise fit algorithm, very scarcely used in climatic trend analyses, mostly based on linear regression. As we show by rigorous statistical analysis, simple linear fit offers a poor description of time changes, compared to non-linear techniques. Whether applying existing algorithms to data is "innovative" or not is just an opinion that we do not share

at all.

-" In what concerns the analysis of results, I generally found it unconvincing," Rew1 states that our statistical analysis is "unconvincing", but gives no statistical analyses or reasons to support that subjective opinion.

- "The recent deceleration of global warming is a global process, not specifically linked to anything peculiar of SE Spain", Rew1 has clearly misunderstood our statements to put in context the interest of using new statistical approaches to time series analysis. In our manuscript we have never linked global deceleration to local SE trends. We simply say that global averaged time series reflect trend changes that can be detected in many local series, as happens with the called "hiatus" from the early 90s. It is out of the scope of this research to discuss about the statistical significance of this hiatus, in global or local series, or about the proposed hypothesis of physical mechanisms. We just mention the existence of a reduction in warming rates throughout the world (as the IPCC last report 5AR does), as a starting point of observations to motivate the use of non-linear approach.

- "Attributing the global deceleration to specific land use changes in a very localized region is certainly wrong" We have never done so. We just speculate in the discussion on the possibility that differences in breakpoints between local series might be due to land use changes, or simple to internal variability. We do not give any definitive conclusion on this respect.

- "I believe that a global analysis of mean or of spatially distributed surface temperature would probably find a breakpoint in the beginning of 2000s" Again, this is a subjective opinion not based on data analyses used to argue against our statistical approach. Visual inspection of data is again just an starting point of observation that can not be used to refute data analysis.

- "he so called "hiatus" seems to be over, after the record breaking temperatures in 2015" Again, a subjective opinion not based on data analysis. As we have said, here

we do not analyze or discuss this global hiatus. Anyway, as this Rew1 must know, one year does not make significant changes in long term statistics.

-"Finally I found the discussion supported by Table 2-4 rather weak. The piecewise approach will always produce a better fit, the question being the statistical support for the extra degrees of freedom."

We chose piecewise regression rather than simple linear regression in an attempt to obtain a better fit as well as to improve the predictive performance. The empirical results with the analyzed data suggest that the choice was appropriate, in our opinion. Certainly, the number of parameters is higher when using piecewise regression and it affects the value of the coefficient of determination. However, the so-called adjusted coefficient of determination, where the effect of the number of parameters is removed, also shows a clear edge for piecewise regression. Furthermore, in cases where linear regression provides a very poor fitting, for instance when analyzing maximum temperatures in Almeria with an adjusted R2=0,00172, the piecewise regression yields a fairly good fitting (adjusted R2=0,4978). We would be happy to substitute R2 by the adjusted R2 if the Referees deem it appropriate. Also, table 2 displays the residual standard error, which is not affected by the differences in number of parameters. These RSE are lower in piecewise regression than linear regression except in minimum temperatures in MA. Hence, we believe the differences in the breakpoints fund in nearby stations are not a side effect of the statistical model used, but they could be instead explained by local forcings of simply natural variability. Though it is out of the scope of our research, we have studied deeply the case of the effect of land use changes in Almeria in other papers reflected in the References section, using both data analysis and numerical simulations (Campra et al, 2008 and Campra and Millstein, 2011)

-" The large differences found in the location of the breakpoints in nearby stations mostly makes me feel uncomfortable about the robustness of the individual results and not excited to look for the physical explanation of those differences." Again, Rew1 gives opinions or feelings not based on statistical analyses. Again, he misunderstands the

scope of our work, not related to give any physical explanations, but solely to offer a better non-lineal trend analysis for climate changes than conventional widespread simple linear analyses.
* * *

---

## Referee Comment (RC2) · Anonymous Referee #2 · 31 Jul 2016

The paper is interesting the authors present the use a nonlinear regression model to analyze the trends of climate data. It is no completely new but there are few Works that use this techniques. In my opinion the paper is too center in the statistical analysis and more explanation about the possible reasons or justifications of these breakpoints in the periods observed are necessary in a climatic sense. Could be the breakpoint observed in 1996 with climate shift? The reasons of the breakpoints should be look for in the possible changes of atmospheric circulation. The authors show some reasons like the Pinatubo eruption but more discussion in this sense is necessary in order to justify the use of this method instead of a simple regression trend. Also would interesting if the authors could say why the method work different in Tmean that in

Tmax or Tmin if they observed the table 2 the differences between piecewise and linear regression methods are less in Tmean.
* * *

---

## Author Comment (AC2) · 26 Sep 2016

Dear Sr, Thanks for carefully reading our paper and for your interesting suggestions on key issues. I answer to your suggestions in between lines.

A) (1) comments from referee 2 and author's responses

1- "In my opinion the paper is too center in the statistical analysis and more explanation about the possible reasons or justifications of these breakpoints in the periods observed are necessary in a climatic sense. . .

Our intention here was just to describe the changes in surface temperature series in the area for a limited period of study. So this study is limited to two goals: -First, by fitting a simple linear regression model, estimate statistically significant trends -Second, an complementary to simple linear fit, by fitting a piecewise regression model, detect and characterize internal and significant breakpoints inside the time series. So, this is just a "detection" study, not an "attribution" study. Here, we just deal with the detection of changes, not with attribution to physical explanations. As stated by the IPCC Good Practice Guidance Paper (Hegerl et al., 2010): Detection of change is defined as the process of demonstrating that climate or a system affected by climate has changed in some defined statistical sense without providing a reason for that change. An identified change is detected in observations if its likelihood of occurrence by chance due to internal variability alone is determined to be small, for example <10%. We have presented here such a detection study, using $p < 0.05$ as significance level. However, our study does not deal at all with attribution, is defined at the same Guidance as "the process of evaluating the relative contributions of multiple causal factors to a change or event with an assignment of statistical confidence. The process of attribution requires the detection of a change in the observed variable or closely associated variables". As you have stated, the most interesting consequence of detecting breakpoints in temperature time series is the search of new forcings of climatic nature that might have driven those changes in linear trends. However, that goal is further away than our intention in this short communication paper. The issue of understanding the sensitivity of the climate system to both natural and anthropogenic changes still remains even in long term temperature changes. This task is much more difficult when trying to explain causality of short or mid-term breakpoints in a local time series, were non-linear climate response might be linked to smooth changes in local forcings. Nevertheless, in the manuscript we have suggested some future lines of research in the field of attribution studies aimed to estimate the impact of dramatic land use changes around two of the stations, MA and AL series. For this, numerical simulations with mesoscale climatic models are the best way to study such attribution to physical forcings, a methodology that is totally out of the scope of this paper. In the case of AL (Almeria), we have indeed carried out these climatic simulations with WRF model and obtained solid evidences of land use changes influence on long term trends (Campra et al, 2008, Campra and Millstein, 2013).

2-"The reasons of the breakpoints should be look for in the possible changes of atmospheric circulation""

As stated above, this kind of attribution link is not possible simply by regression studies as ours. As Tome and Miranda (2004) state, turning points obtained by piecewise regression are only related to a given series of data used for the fit. This way, for instance, it must be beared in mind that breakpoints obtained for global temperature series might not have special meaning for other local or regional climate series.

3-"" more discussion in this sense is necessary in order to justify the use of this method instead of a simple regression trend."

As we state in the introduction section, our initial justification for the use of non-linear fitting regression instead of conventional simple linear regression was the possibility to detect inner changes in long term climatic trends and their structural changes. We have chosen piecewise regression in an attempt to obtain a better fit as well as to improve the predictive performance of the model. Our starting point was the few previous works were piecewise regression was applied to temperature series (Karl et al, 2000; Tome & Miranda, 2004). In none of these attribution analysis are presented, due to the issues explained above. Furthermore, at the discussion and conclusion we have shown that the empirical results with the analyzed data suggest that the choice was appropriate, in our opinion. This model allowed us to better describe the inner structure of change and determine breakpoints in long term trends in surface air temperatures at the main SE Spain records. In conclusion, we present here a non-linear fit COMPLEMENTARY to, but not instead of a simple linear fit. We have explained this idea with a new paragraph on p34, L9.

4-" Could be the breakpoint observed in 1996 with climate shift?"

We guess that you mean if we have detected breakpoints around the global climate cooling caused by the Pinatubo eruption in 1991. Though this disturbance can be clearly seen by causal inspection of time series (Fig 1), it was a short-term cooling and global temperatures recovered their mean values in 1-2 years. The breakpoints are detected by checking if there exist a significant difference between the slopes from both segments with the restriction of each segment includes a time period of, at least, 5 years. So, the non-detection of the previously mentioned point can be due to the restriction or to the lack of significance in the change of the slopes. Due to this restriction given to the piecewise algorithm, it does not yield a breakpoint exactly at 1991-92, but in general breakpoints are located around the early 90s for every series. To avoid confusion about this point, we have now deleted the mention to Pinatubo eruption in the reviewed text, taking into account that we have not studied the attribution of our breakpoints.

5- "the authors could say why the method work different in Tmean that in Tmax or Tmin"

It is logical that there is less difference in the goodness of fit for Tmean, as the time series is obtained by averaging Tmin and Tmax, obtaining a smoothed time series.

We have included this idea in the text (P7, L10)

  b) Changes in the manuscript

In order to make clearer these issues, and improve the manuscript with the suggestions of Referee 2, we present a more conservative manuscript, by deleting or modifying the text were the main doubts or confusion have been pointed out by referees 1 and 2 , basically dealing with attribution of changes or the global pause issue. We have made these changes in the manuscript:

P4 L15. Deleted the reference to global pause ("However, recent warming trends throughout the planet. . .)

P4 L30 . End of introduction section. Deleted last lines (p4, l30-37) and inserted new lines to make clear that we present a "detection", not an "attribution study" (p5, L1 y ss, "Here we present. . ."

P7, L10. Inserted explanation of differences between Tmean vs Tmin and Tmax at table 2.

P7, L17. Mention to Pinatubo eruption deleted.

P9, L10. Last line deleted from, "Our regression analyses. . ."

P9, L24. Deleted from "no matter the uncertantiy of breakpoints. . ."

P9, L29. Deleted from " Furthermore, recent breakpoints observed in MA and GR..",

P9, L37. Deleted from "To obtain statistical evidence. . ."

P10, L3. Inserted final suggestion on attribution studies: "Future attribution studies. . .

P10, L6. Inserted ", neither that long term warming trends in the area of study have come to an end."

P10, L9. Deleted final comment "Complementary to this search of statistical relationships. . ."

P34, L9. Inserted a final conclusion justifying the use of piecewise fit, "complementary" to simple linear approach: "We believe that our analysis of piecewise regression fit. . ."

Finally, references included at the deleted text have been deleted, and a new reference has been added:

Hegerl, G.C., O. Hoegh-Guldberg, G. Casassa, M.P. Hoerling, R.S. Kovats, C. Parmesan, D.W. Pierce, P.A. Stott, 2010: Good Practice Guidance Paper on Detection and Attribution Related to Anthropogenic Climate Change. In: Meeting Report of the Intergovernmental Panel on Climate Change Expert Meeting on Detection and Attribution of Anthropogenic Climate Change [Stocker, T.F., C.B. Field, D. Qin, V. Barros, G.-K. Plattner, M. Tignor, P.M. Midgley, and K.L. Ebi (eds.)]. IPCC Working Group I Technical

Support Unit, University of Bern, Bern, Switzerland.

We enclose here the reviewed manuscript with tracked changes.

Regards,

P Campra M Morales

Please also note the supplement to this comment:
http://www.nonlin-processes-geophys-discuss.net/npg-2016-29/npg-2016-29-AC2-supplement.pdf

[Figure]

**Supplement:**

[revised manuscript text omitted]
.  Here we present a statistical characterization of the recent changes in surface temperatures in this area, estimating both simple linear trends and main short-term inner changes of terms inside the period of study. Though we suggest some possible physical explanations for some breakpoints, here we do not intend to present an attribution analysis of the variability of the time series, though. Our study is limited to the detection and descriptions of the patterns of change by regression analyses. In this sense, our objectives were:

1. Update temperature trends in SE Spain, based on the records of the main reliable stations and using conventional linear regression fit.

2. Further describe the inner structure of temperature changes by fitting a nonlinear model (piecewise regression)

**2. Data temperature series**

We have selected four meteorological stations with the longest, continuous and most reliable records in SE Spain (Table 1): Almeria (AL), Granada (GR), Malaga (MA), and Murcia-San Javier (MU). These stations are well-spaced across SE Spain (Fig. S1), with a minimum linear distance of 90 Km (between GR and MA), and maximum of 340 km between MA and MU, below the threshold of 400 km that has been suggested as optimal for building a representative meteorological network in Spain (Peña-Angulo et al., 2014). AL, MA and MU are representative of coastal Mediterranean climate, while GR shows a "continentalized" inland Mediterranean climate, with higher extremes of warm and cold days in summer and winter, respectively. Their records cover at least from 1973 to 2014, including the recent period of accelerated global warming from the 1970s that we intended to analyze in our area of study. These stations are located within international airports, and belong to the first order (synoptic) network of the Spanish official meteorological agency (AEMET). This selection was based on potential data quality from highly monitored sites and good-quality records controlled by the *Servicio de Desarrollos Climatológicos* (SDC, Climatological Branch) of AEMET. Raw data of daily $T_{max}$ and $T_{min}$ have undergone quality control checks to avoid syntax, internal consistency and temporal coherence errors, and controls of extreme thresholds and spatial coherency. Additionally, these records have been extensively analyzed for artificial in-homogeneities in previous studies (Brunet et al., 2008; Staudt et al., 2007).

**3. Regression methods**

**3.1. Simple linear regression**

As a first conventional approach to detect temperature change, simple linear regression was applied to the series. Trends and their 95% CIs were estimated by least squares linear regression. Linear trends were estimated in every series from the slopes of the fit using values of annual averages of $T_{max}$, $T_{min}$, $T_{mean}$ and DTR, calculated from monthly means provided by AEMET.

**3.2. Piecewise regression**

However, when the residuals of simple linear fits for each $T_{mean}$ series were tested for the assumptions of normality, independence, homoscedasticity and linearity, it turned out that: a) homoscedasticity was is not met by GR station; b) the linear assumption was not verified by AL and MU residuals; c) the independence assumption was rejected by Ljun-Box in AL. The violation of the homogeneous variance assumption could result in unreliable estimates of the standard errors that might turn out in mistaken conclusions over the slope. In these cases, heteroscedasticity-consistent (MacKinnon and White, 1985), and autocorrelation-consistent estimators have been used (Newey and West, 1987). To solve these problems, here we have tested and alternative regression model, piecewise regression. We have used a segmented model between the mean response E(Y) and the explanatory variable Z, modeled by adding in the linear predictor the terms. Eq. (1):

$$\beta_1 z_i + \beta_2 (z_i - \psi)_+ \qquad (1)$$

where $\beta_1$ is the slope of the left line segmented, $\beta_2$ is the 'difference-in-slopes', $\psi$ is the breakpoint, and $(z_i - \psi)_+ = (z_i - \psi) \times I(z_i > \psi)$ being I(A)=1 if A is true. In order to estimate the break-points location, we use the approach suggested by Muggeo (2003) and at the R package 'segmented' (Muggeo, 2008). Karl *et al.* (2000) used this approach to obtain a better fit of global temperatures than simple linear regression. Tomé and Miranda (2004) further developed an algorithm to identify best location for breakpoints in climatic series. Here we have applied a piecewise regression model to those series where there is enough statistical evidence to support the existence of breakpoints. Smoothed scatter plots were used to provide the starting values for breakpoints in order to improve the convergence of the algorithm, and we have checked the existence of a significant breakpoint by testing over the difference in slope (Muggeo, 2003). We have employed the R package to estimate the parameters of the piecewise regression in a deterministic way, and to fit linear segments to the data. The analysis of the residuals from piecewise regression fits of our data showed that the assumptions of normality (Shapiro-Wilks and Anderson-Darling tests), independence (Ljung-Box test) and homocedasticity (Breusch-Pagan test) were met by all the series, with the exception of GR. In this case, the robust variance estimator proposed by MacKinnon and White (1985) was used. In order to test for a significant slope, we have applied a Wald's test. As general criteria, we have not estimated the slope of segments when they represented time intervals of less than 5 years. As stated by Tome and Miranda (2005), ", if the first or the last breakpoint happen near the minimum allowed position the result should be looked upon with some suspicion". These graphs are not shown in the figures at the results section. hh

**3.3. Evaluation of regression models**

We have compared both fitting and predicting performance of piecewise and simple linear models applied to the whole length of data available for every series of $T_{mean}$, $T_{max}$ , $T_{min}$ and DTR (Table 2). In order to compare the goodness-of-fit of the models, we calculated $R^2$ for each fit, and the residual standard error (RSE) in order to avoid the artificial skill of $R^2$. Also, we carried out a cross-validation analysis (Hastie et al., 2009) to compare the goodness-of-forecasting skills among models. Thus, the data were divided into 5 roughly equal-sized parts, and for the $i_{th}$ part, i=1,…,5, the model was fitted using the other 4 parts of the data. The prediction error of the fitted model was estimated when predicting the $i_{th}$ part of the data. Finally, the mean square error ($MSE_{CV}$) of the 5 evaluation parts was calculated as forecasting skill indicator. As we can observe in Table 2, piecewise regression showed a superior behavior both fitting and forecasting compared to simple linear regression. It must also be noted that, due to smoothing by averaging extreme temperature values, the differences between piecewise and linear regression methods are less in $T_{mean}$ thant in $T_{max}$ or $T_{min}$.

In general, piecewise regression allowed obtaining a better fit and an improved predictive performance than simple linear model. Though it can be argued that $R^2$ values are affected by a higher number of parameters used for piecewise model, the values of adjusted $R^2$, where the effect of the number of parameters is removed, also confirmed the better fit for piecewise regression. Furthermore, in cases where linear regression provided a very poor fitting, as with $T_{max}$ in AL adjusted $R^2$=0.001, the piecewise regression yields a fairly good fitting (adjusted $R^2$=0.49). Better performance is also characterized by lower residual standard error (RSE), which is not affected by the differences in number of parameters. These RSE are lower in piecewise regression than linear regression (Table 2) (except in $T_{min}$ for MA). Hence, we believe the differences in the breakpoints fund in nearby stations are not a side effect of the statistical model used, but they could be instead explained by either undetermined local forcings or internal variability.

**4. Results and discussion**

[revised manuscript text omitted]
. We believe that our analysis of piecewise regression fit shown here justifies the suitability of the employment non-linear models in climatic change studies. Results obtained with our piecewise regression model showed that it can be a useful complement to the conventional simple linear models for a more detailed description of changes in climatic variables. By flexible regression, we have detected a recent slow-down in long-term warming trends in three of the four main records in SE Spain, with no significant trends in $T_{mean}$ in the most recent segments of two of them (AL and GR)

exact location of breakpoints reported here, given by 95% CIs (Table 4), we have shown statistical evidence of the presence of such reduction in warming rates in the area, consistent with previous reports of recent decreasing warming trends in global series. However, and given the limited length of the observations, we cannot suggest by our analysis that the absence of statistically significant warming in the most recent segments of piecewise regression model are necessarily inconsistent with historic variability of the series studied. Also it is important to notice that by simply using both regression analysis, linear and piecewise, we cannot make projections of future trends, neither suggest that reduced linear trends in the area of study continue in the next years . Furthermore, recent breakpoints observed in MA and GR, might be signing increasing warming trends in the future, enhanced by 2014 global historic record (Jones, 2015), year-to-date global temperatures (up to June 2015) (NOAA, 2015a). Spatial differences in long-term trends and breakpoints location might be due to undetermined local dynamics in climate and/or forcings that still remain to be determined in every station, or just simply due to natural variability. It must also be taken into account that the method of piecewise regression is strongly dependent on the first guesses and imposed parameters such as minimum time distance between breakpoints, and minimum trend change between breakpoints (Tome and Miranda, 2005). Future attribution studies, such as those recommended at IPCC Good Practices Guidance (Hegerl et al., 2010), including numerical simulations with mesoscale models and extended local forcings might help explaining the observed variability, and the location of the trend changing points in the temperature series described here. Complementary to this search of statistical relationships, recent simulation studies with optimized treatment of global climate forcings have shown better insight to possible causes of a possible slow-down in global warming (Smith *et al*., 2014).

*Author contribution.* P. Campra designed the scope of the research, performed the simple linear regressions and wrote the manuscript, and M. Morales carried out all deep statistical analyses, residuals tests and developed the piecewise regression model in R.

*Acknowledgements:* We thank the Ministry of Economy and Competitiveness (MINECO) of the Spanish Government, and FEDER Funds from European Union for the Grant CGL2013-46873-R that has supported this work. The authors would like to acknowledge AEMET (Spanish Meteorological Agency) for providing data used in this work.

**Supporting Information**

Map of location of stations in SE Spain, picture of greenhouses land cover around AL station, picture of MA airport new taxiways. Tables of uncertainty estimates of piecewise regression of mean, maximum, minimum and DTR temperature series. This material is available free of charge via the Internet at …

[revised manuscript text omitted]